# Numerical Investigations for Vibration and Deformation of Power Transformer Windings under Short-Circuit Condition

**Jiawei Wang \*, Yijing Xing, Xikui Ma, Zhiwei Zhao and Lihui Yang**

State Key Laboratory of Electrical Insulation and Power Equipment, School of Electrical Engineering, Xi'an Jiaotong University, Xi'an 710049, China; yijing_xing1@stu.xjtu.edu.cn (Y.X.); maxikui@mail.xjtu.edu.cn (X.M.); zhaozhiwei_2019@163.com (Z.Z.); lihui.yang@mail.xjtu.edu.cn (L.Y.)
\* Correspondence: jwwang@xjtu.edu.cn; Tel.: +86-029-8266-8629

**Abstract:** The analysis of the dynamic process of winding destabilization under sudden short-circuit conditions is of great importance to accurately assess the short-circuit resistance of power transformers. Based on magneto-solid coupling, an axisymmetric model of the transformer and a 3D multilayer model of the transformer considering the support components are established, respectively, and the short-circuit electromagnetic force (EF) is simulated by using the finite element method. It is concluded that the middle layer of the winding is subjected to the larger radial EF, while the axial EF has a greater effect on the layers at both ends. Moreover, the impression of the preload force, aging temperatures, and the area share of spacers on the vibration and deformation of windings are studied under short-circuit conditions. The overall distribution of plastic strain and residual stress in the winding is symmetrical, and the maximum values occur in the lower region of the middle of the winding. Finally, considering the material properties of disks and insulating components, the cumulative effect of plastic deformation under multiple successive short-circuit shocks is calculated. Compared with the traditional axisymmetric model of transformer, the three-dimensional multilayer model of the transformer established in this paper is more suitable for the actual winding structure and the obtained results are more accurate.

**Keywords:** power transformer; finite element model; winding vibration; cumulative effect

## 1. Introduction

With the improvement in the voltage level of power systems, short-circuit accidents caused by the damage of the transformer have been becoming more and more serious [1,2]. The stability and safety of power transformers determines the reliability of the power system. When a short-circuit fault occurs, the short-circuit current is approximately dozens of times the rated current, which leads to a short-circuit electromagnetic force (EF) hundreds of times stronger than that under normal operating conditions [3]. Under such strong EF, transformer windings are highly susceptible to accidents such as deformation, inter-turn insulation breakage, and instability.

We refer to the coupling between the EF on the winding and its structural deformation as magneto-mechanical coupling. A considerable number of researchers have developed transformer models using the finite element method to study the magneto-structure coupling effect of windings [4–7]. The researchers equated transformer disks as concentrated masses and insulation spacers as springs to study the axial vibration of the windings based on the mass–spring model [8,9]. After a short-circuit shock in the transformer, the winding deformation affects both the current distribution and the leakage flux density distribution, thus changing the EF acting on the winding, which in turn affects the winding deformation. Recently, many research results have also shown that the deformation and structure of windings have a non-negligible effect on EF [10–13].

The main factors affecting the vibration and deformation of transformer windings include the nonlinearity of the insulation spacer material [14] and the axial preload force. In

our work, magneto-mechanical coupling models of the power transformer are established to investigate the distribution of leakage flux and short-circuit EF as well as factors of the winding vibration. In addition, the cumulative effect of plastic deformation under multiple successive short-circuit shocks is analyzed.

## 2. Theory and Formulations

### 2.1. Electromagnetic Analysis

The completion of electromagnetic analysis is twofold. Firstly, the short-circuit current is extracted based on a circuit model of transformers. Secondly, the short-circuit current is imposed to the windings as the excitation of Maxwell's equation to calculate the detailed distributions of both leak flux and electromagnetic force.

It is worth emphasizing that when short-circuit faults occur, the cores of the transformers are not saturated [15]; thus, we can assume a constant short-circuit impedance and start with the equivalent model shown in Figure 1, which is governed by

$$u_1 = \sqrt{2}U_1 \sin(\omega t + \alpha) = i_k R_k + L_k \frac{\mathrm{d}i_k}{\mathrm{d}t},\tag{1}$$

where $u_1$ is the primary voltage, $U_1$ is the root mean square (RMS) of $u_1$, $\omega$ is the angular frequency, $t$ is the time, $\alpha$ is the initial phase angle, $i_k$ is the transient short-circuit current, and $R_k$ and $L_k$ are the short-circuit resistance and inductance, respectively.

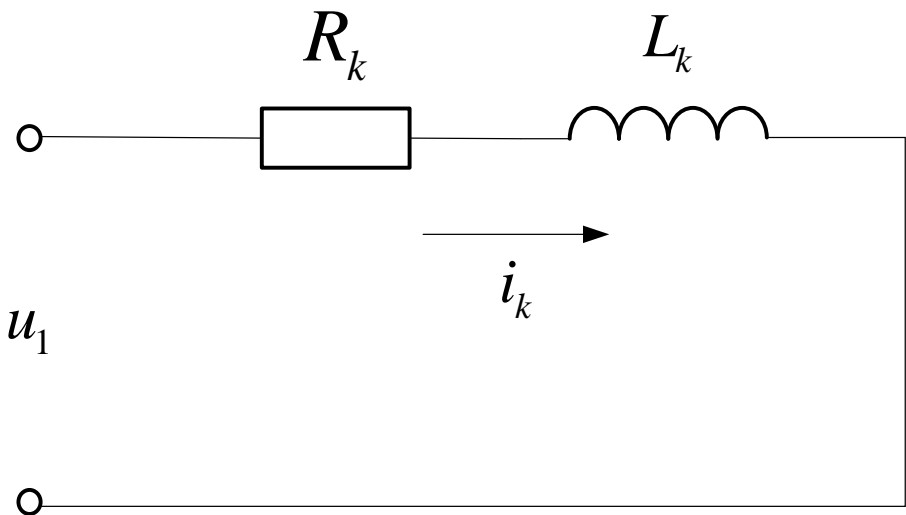

**Figure 1.** Equivalent model of transformers under short-circuit condition.

The transient short-circuit current $i_k(t)$ can then be calculated as

$$i_k(t) = \sqrt{2}I_k[\cos \alpha e^{-\frac{R_k}{L_k}t} - \cos(\omega t + \alpha)],\tag{2}$$

where $I_k$ is the RMS of steady-state current.

Once $i_k(t)$ is obtained, it can be used to excite the Magnetoquasistatic (MQS) field governed by

$$\begin{cases} \mathbf{E} = -\frac{\partial \mathbf{A}}{\partial t} \\ \nabla \times \mathbf{H} = \mathbf{J} \\ \nabla \cdot \mathbf{A} = 0 \\ \mathbf{B} = \nabla \times \mathbf{A} \end{cases}\tag{3}$$

as well as the constitutive law

$$\begin{cases} \mathbf{J} = \mathbf{J}_e + \sigma(\mathbf{E} + \mathbf{v} \times \mathbf{B}) \\ \mathbf{B} = \mu \mathbf{H} \end{cases}.\tag{4}$$

In the above equations, $\mathbf{A}$ is the magnetic vector potential, $\mathbf{E}$ is the electric field intensity, $\mathbf{H}$ is the magnetic field intensity, $\mathbf{B}$ is the magnetic flux density, $\mathbf{J}$ is the current density, $\mathbf{J}_e$ is the applied current density, $\mathbf{v}$ is the velocity of the conductor, $\sigma$ is the electrical conductivity, and $\mu$ is the magnetic permeability. A pregiven $\mathbf{J}_e$ is imposed on the ports of windings.

After the extraction of the leakage flux via solving Equations (3) and (4), the resultant volume density of the short-circuit EF is given by

$$\mathbf{F}_e = \mathbf{J} \times \mathbf{B}. \tag{5}$$

### 2.2. Mechanical Analysis

For accurate analysis of the elastic–plastic deformation and the cumulative effect of transformer winding, it is necessary to determine the internal stress state induced by EF on the windings. The short-circuit EF is a time-varying and uneven physical force exciting the transient structural equation

$$\rho \frac{\partial^2 \mathbf{u}}{\partial t^2} = \nabla \cdot \boldsymbol{\sigma} + \mathbf{F}_e, \tag{6}$$

where

$$\boldsymbol{\sigma} = \begin{bmatrix} \sigma_{xx} & \tau_{xy} & \tau_{xz} \\ \tau_{yx} & \sigma_{yy} & \tau_{yz} \\ \tau_{zx} & \tau_{zy} & \sigma_{zz} \end{bmatrix}, \tag{7}$$

is the stress tensor $\rho$ is the mass density, $\mathbf{u} = [u_x, u_y, u_z]^{\mathrm{T}}$ is the displacement vector. It is worth noting that in Equation (7) $\tau_{ij} = \tau_{ji}$, where $i, j \in \{x, y, z\}$. Similarly, the strain tensor

$$\boldsymbol{\varepsilon} = \begin{bmatrix} \varepsilon_{xx} & \gamma_{xy} & \gamma_{xz} \\ \gamma_{yx} & \varepsilon_{yy} & \gamma_{yz} \\ \gamma_{zx} & \gamma_{zy} & \varepsilon_{zz} \end{bmatrix} \tag{8}$$

is also a symmetric tensor. The two tensors are linked through the constitutive laws

$$\begin{bmatrix} \sigma_{xx} \\ \sigma_{yy} \\ \sigma_{zz} \end{bmatrix} = \frac{E}{(1+v)(1-2v)} \begin{bmatrix} 1-v & v & v \\ v & 1-v & v \\ v & v & 1-v \end{bmatrix} \begin{bmatrix} \varepsilon_{xx} \\ \varepsilon_{yy} \\ \varepsilon_{zz} \end{bmatrix}, \tag{9}$$

and

$$\begin{bmatrix} \tau_{xy} \\ \tau_{yz} \\ \tau_{zx} \end{bmatrix} = \frac{E}{2(1+v)} \begin{bmatrix} \gamma_{xy} \\ \gamma_{yz} \\ \gamma_{zx} \end{bmatrix}, \tag{10}$$

where $E$ and $v$ denote the elasticity modulus and Poisson's ratio, respectively.

The displacement vector $\mathbf{u}$ is related to $\boldsymbol{\varepsilon}$ by the geometric law

$$\boldsymbol{\varepsilon} = \frac{1}{2} \left[ (\nabla \mathbf{u})^{\mathrm{T}} + \nabla \mathbf{u} \right]. \tag{11}$$

### 2.3. Finite Element Solution to the Coupled Magneto-Mechanical System

The finite element method (FEM) is adopted for the numerical solution to the coupled magneto-mechanical equations due to its capability of resolving complex geometry. The resultant discretized system in a compact matrix form can be described by

$$\begin{pmatrix} \mathbf{M} & 0 \\ 0 & 0 \end{pmatrix} \begin{pmatrix} \ddot{\mathbf{U}} \\ \ddot{\mathbf{A}} \end{pmatrix} + \begin{pmatrix} \mathbf{C} & 0 \\ 0 & 0 \end{pmatrix} \begin{pmatrix} \dot{\mathbf{U}} \\ \dot{\mathbf{A}} \end{pmatrix} + \begin{pmatrix} \mathbf{K}(\mathbf{U}) & 0 \\ 0 & K^m \end{pmatrix} \begin{pmatrix} \mathbf{U} \\ \mathbf{A} \end{pmatrix} = \begin{pmatrix} \mathbf{F}(\mathbf{A}) \\ \mathbf{J} \end{pmatrix}, \tag{12}$$

where $\mathbf{M}$ is the mass matrix, $\mathbf{C}$ is the damping matrix, $\mathbf{K}(\mathbf{U})$ is the stiffness matrix depending on the displacement vector $\mathbf{U}$, $\mathbf{F}(\mathbf{A})$ is the EF force vector calculated according

to the magnetic vector potential **A**, and $\mathbf{K}^m$ is the coefficient matrix representing the MQS equation.

Note that the dependence of $\mathbf{K}(\mathbf{U})$ on **U** makes Equation (12) a nonlinear system. The nonlinearity stems from nonlinear mechanical characteristics of some specific structures, e.g., the insulating spacers. This nonlinearity will be defined in detail in Section 3.

Another point is that although Equation (3) involves the first-order time derivative of **E**, the discretized MQS equation does not contain any time derivative, because we choose to solve only the magnetic vector potential **A**, which behaves like a static field under the MQS assumption.

The FEM discretization of the governing equations is realized with the commercial FEM solver COMSOL Multiphysics 6.0, and we choose the built-in backward Euler algorithm allowing adaptive time step sizes to complete the time advances of Equation (12).

As for the implementation details of the FEM discretization, including the shape functions, evaluation of the coefficient matrices, and time-stepping algorithms, the reader can refer to [16].

## 3. Investigation into an Oil-Immersed-Type 110 kV Power Transformer

### 3.1. Description of the Model

In this section, the vibration and deformation of the windings under the short-circuit condition of a three-phase oil-immersed-type 110 kV power transformer are thoroughly investigated as an example. The transformer has three groups of windings, the rated voltages of which are 110 kV, 38.5 kV, and 10.5 kV. In this work, we focus on the interactions of the latter two groups of windings, referred to as the HV winding and LV winding, respectively. The HV windings are under a star arrangement, while the LV windings are delta-connected. Some key parameters of the transformer are given in Table 1.

**Table 1.** Specification of Oil-Immersed-Type 110 Kv Power Transformer.

| Quantity | Value | Unit |
|---|---|---|
| Rated Power | 63 | MVA |
| Impedance | 6.35 | % |
| Phase voltage of LV/HV windings | 10.5/38.5 | kV |
| Phase current of LV/HV windings | 2000/983.1 | A |
| Frequency | 50 | Hz |
| No. winding turns of LV/HV windings | 76/155 | - |
| No. disks in windings of LV/HV | 76/86 | - |
| Height of each copper disk of LV/HV | 15.55/11.37 | mm |
| Height of window | 1650 | mm |
| Radius of core cross-section | 359 | mm |
| Inner/Outer radius of LV copper disk | 372/431 | mm |
| Inner/Outer radius of HV copper disk | 453/545/5 | mm |
| Initial height of spacer of LV/HV | 3.52/4.41 | mm |

An axisymmetric model of the transformer based on the spring–mass system and a 3D multilayer model of the transformer considering the support components are established, as shown in Figure 2, respectively. The force between the adjacent windings of the three-phase transformer is neglected under a sufficient axial preload, thus we only calculate one phase winding [17]. The electromagnetic field and mechanics share the same set of meshes and have an identical form of mesh division. There are, of course, some similarities and differences between the two finite element models in Figure 1.

In Figure 2a, a simplified axisymmetric model of the transformer is presented, which is obtained by neglecting some unsymmetric geometry details. The axisymmetric model is favorable regarding computational efficiency and suitable for the simulation of the axial vibration of the winding. In Figure 2b, the 3D multilayer model reflects the actual spacers' distribution and fits better to the real winding structure. Meanwhile, the number of turns

of primary and secondary windings remains the same with the parameters in Table 1, and the number of disks is simplified to some extent to save calculation time.

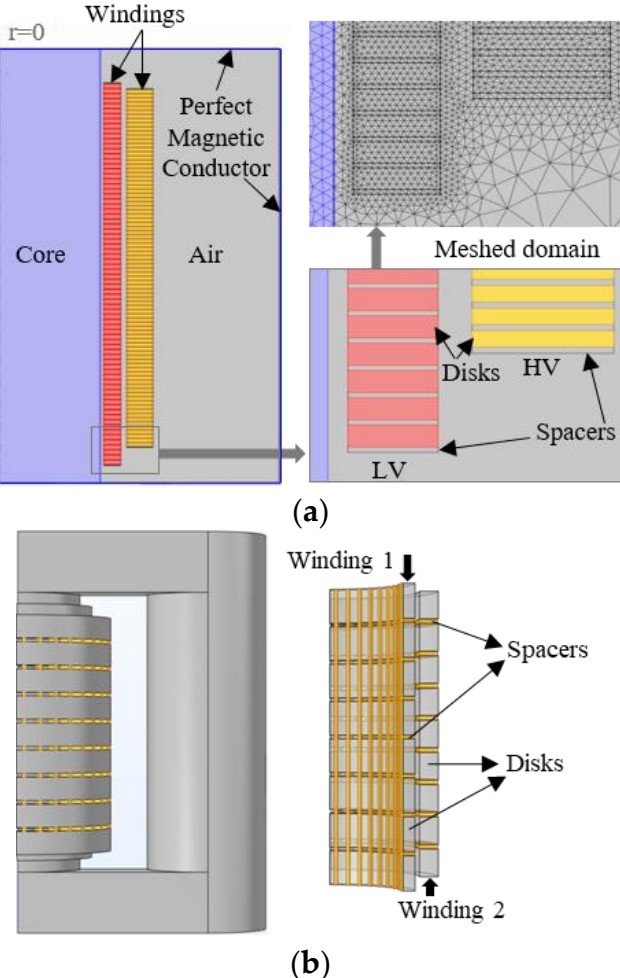

**Figure 2.** Finite element model of the transformer: (**a**) axisymmetric model and (**b**) 3D multilayer model.

As for material properties, the core is made of soft iron with a nonlinear B-H curve, which is depicted in Figure 3. The windings are made of copper with $\sigma = 5.7 \times 10^7$ S/m, $\mu_r = 1$, $E = 110 \times 10^9$ Pa, and $v = 0.34$. For disks and insulating spacers, we set $\sigma = 1$ S/m, $\mu_r = 1$, and $v = 0.30$. The nonlinear elastic modulus is specified in Section 3.4.1.

### 3.2. Calculation of the Leakage Flux and Short-Circuit EF

Here, the short-circuit current is considered when a three-phase symmetric short-circuit fault occurs. After a short circuit occurs, the peak short-circuit current reaches its maximum, which is approximately $7 \times 10^4$ A, at $t = 0.01$ s. The transient short-circuit current obtained by Equation (2) is shown in Figure 4, and it is applied to the winding as an excitation source.

The leakage flux at the time instant $t = 0.01$ s after the sudden short circuit, i.e., when the instantaneous value of the short-circuit current reaches its maximum, is shown in Figure 5. The short-circuit EF is affected by both the short-circuit current and the leakage field. The distributions of EF in HV and LV windings at 0.01 s are given in Figure 6. It is worth noting that in our 3D simulation, the number of tetrahedral elements is 164443, which is sufficient to yield almost grid-independent solutions for our example. Therefore, the simulation results here and in the following sections are reliable from the numerical perspective.

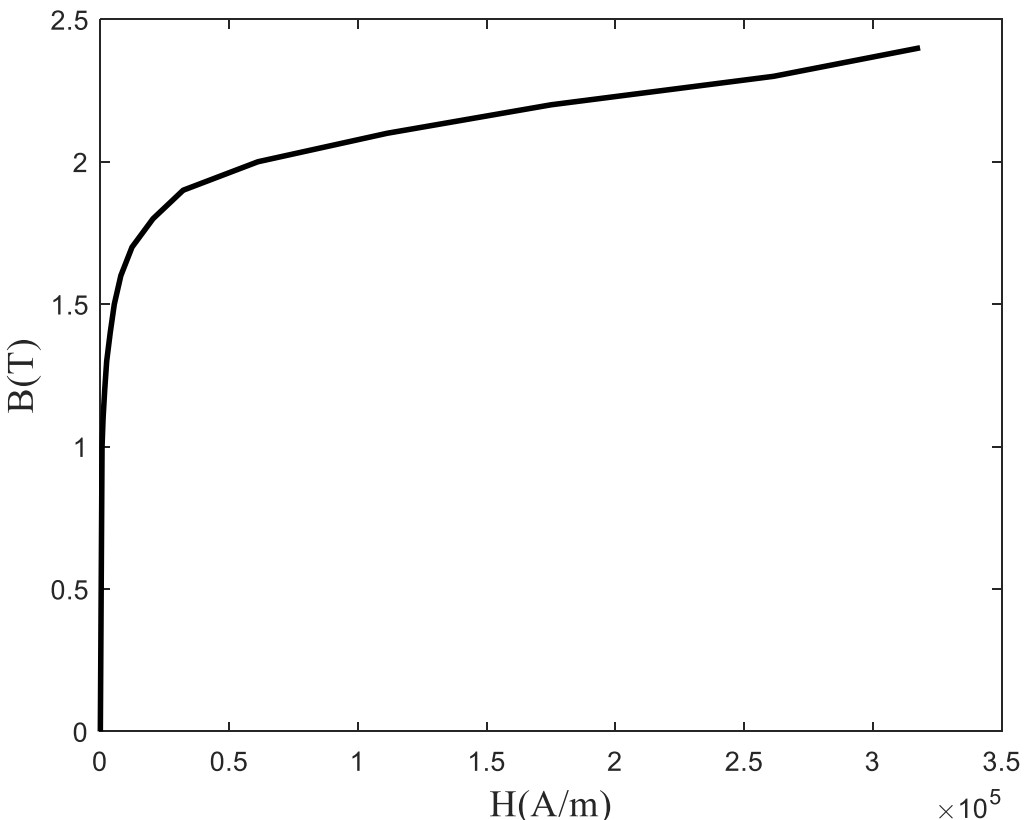

**Figure 3.** B-H curve of the core.

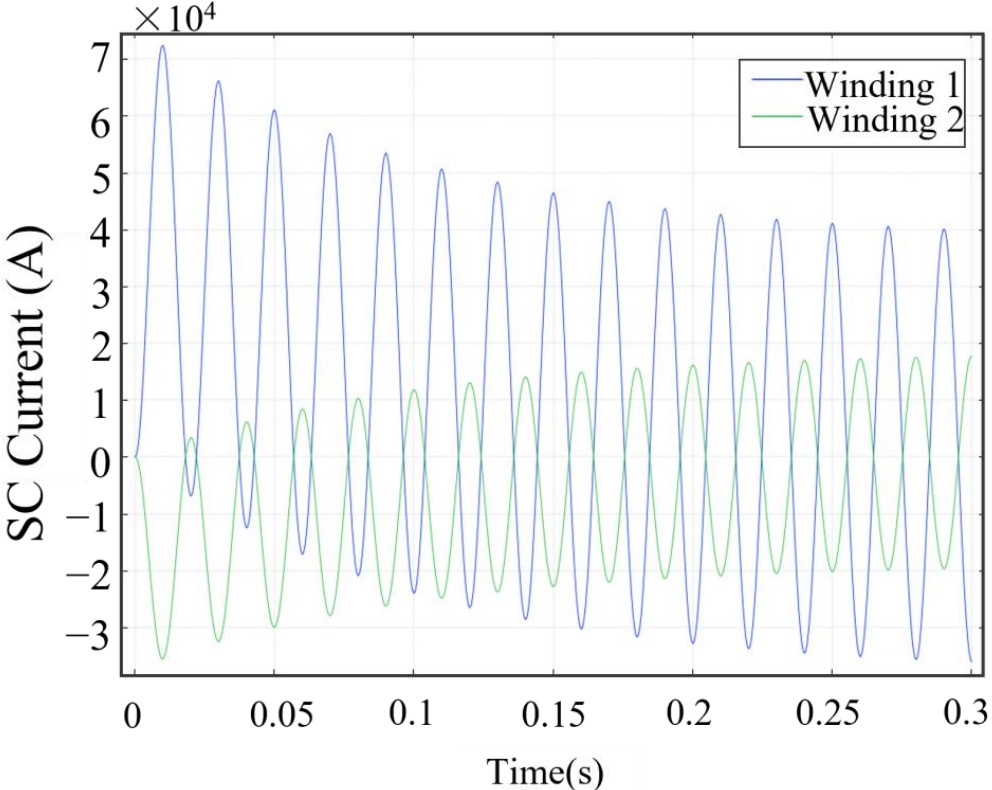

**Figure 4.** Transient short-circuit current in different windings.

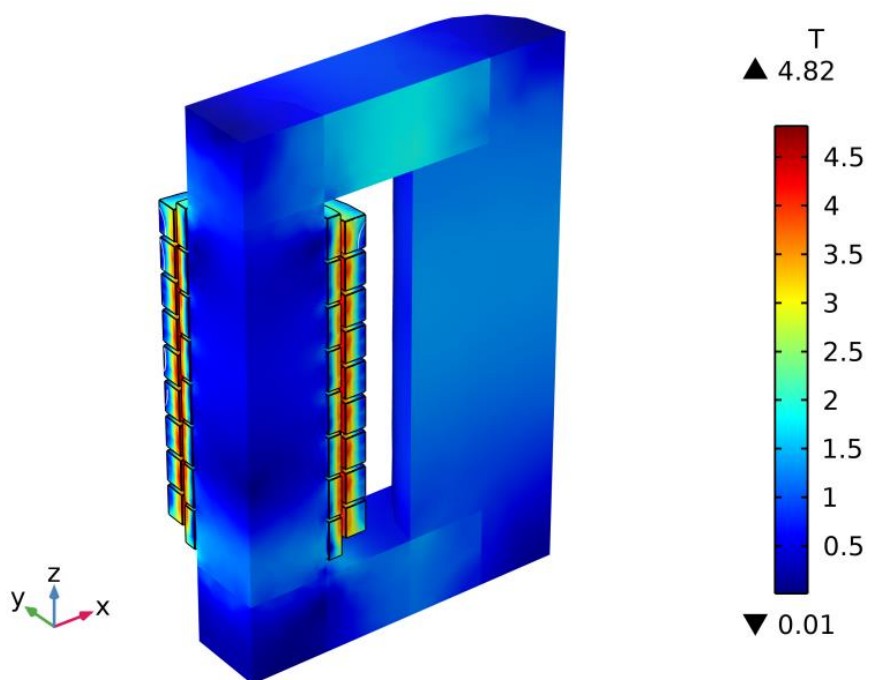

**Figure 5.** Leakage flux of transformer windings under fault currents.

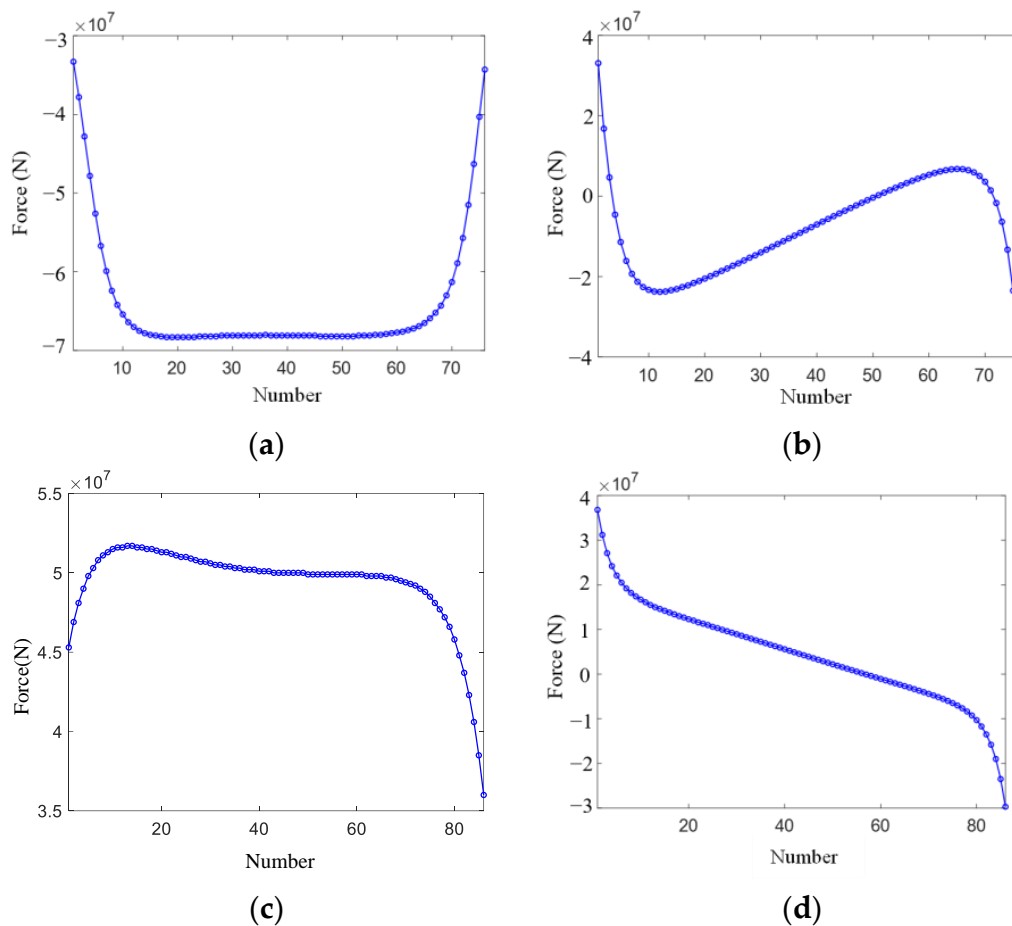

**Figure 6.** Short-circuit EF applied to the center of each winding: (**a**) radial component of EF in LV winding, (**b**) axial component of EF in LV winding, (**c**) radial component of EF in HV winding, and (**d**) axial component of EF in HV winding.

The overall radial component of winding EF is strong in the middle and mild at both ends, due to the concentrated distribution of axial leakage in the middle. The negative radial component of the short-circuit EF in the primary winding means that the primary winding is compressed inward, while the positive radial component of the short-circuit EF in the secondary winding indicates that the secondary winding is expanded outward. As the radial leakage is concentrated at both ends, the axial EF is at its maximum at the end of the winding. In addition, the closer it is to the middle of the winding, the smaller the axial force is.

### 3.3. Dynamic Analysis of Winding Vibration and Deformation

#### 3.3.1. Effect of the Area Proportion of Spacers on Axial Vibration

The contact area between the disk and the spacer is an essential indicator for the design of winding short-circuit resistance. Because all insulation spacer layers have the same structure, only one of them is illustrated. Figure 7 shows a schematic cross-section of a particular insulating spacer layer along the axial direction, where the white area is the air and the gray area indicates the insulating pads. By varying the width of each spacer to change the contact area between each layer of the disks and the insulating spacers, the study index is set to the percentage $\varphi_1$ of the spacer contact area over the circumference of the spacers.

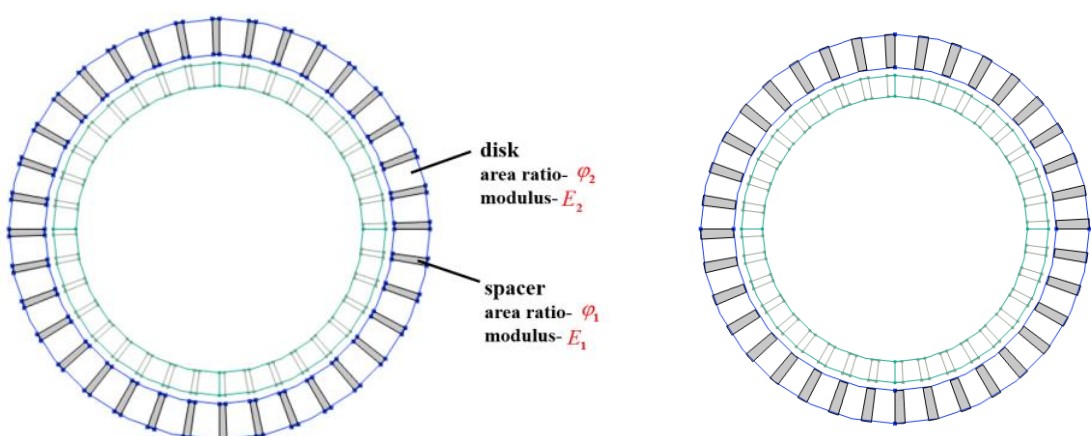

**Figure 7.** Different contact areas between disks and spacers.

Considering the insulating spacer layer as a composite material consisting of air and insulating paperboard, the composite elasticity modulus can be expressed as:

$$E_\varphi = \varphi_1 E_1 + \varphi_2 E_2 \tag{13}$$

where $\varphi_1$ is the area proportion of spacers in each layer, $E_1$ is the elasticity modulus of the cushion block material, $\varphi_2$ is the area proportion of air in each layer, and $E_2$ is the elasticity modulus of air. If the contact areas of the spacers and the wire cakes vary, $\varphi_1$, $\varphi_2$, and the resultant equivalent elasticity modulus of the winding layers will change accordingly.

In Figure 8, the horizontal coordinates indicate the numbering of the studied object. The neighboring biscuits are numbered in order from top to bottom. For instance, No.1 indicates that the studied objects are the two biscuits adjacent to the uppermost end of the winding, and No.2 represents that the studied objects are the second and third biscuits counted from the upper end. The LV winding has a total of 76 bobbins, so there are 75 numberings in total. The HV winding has a total of 86 bobbins, resulting in a total of 85 numbers.

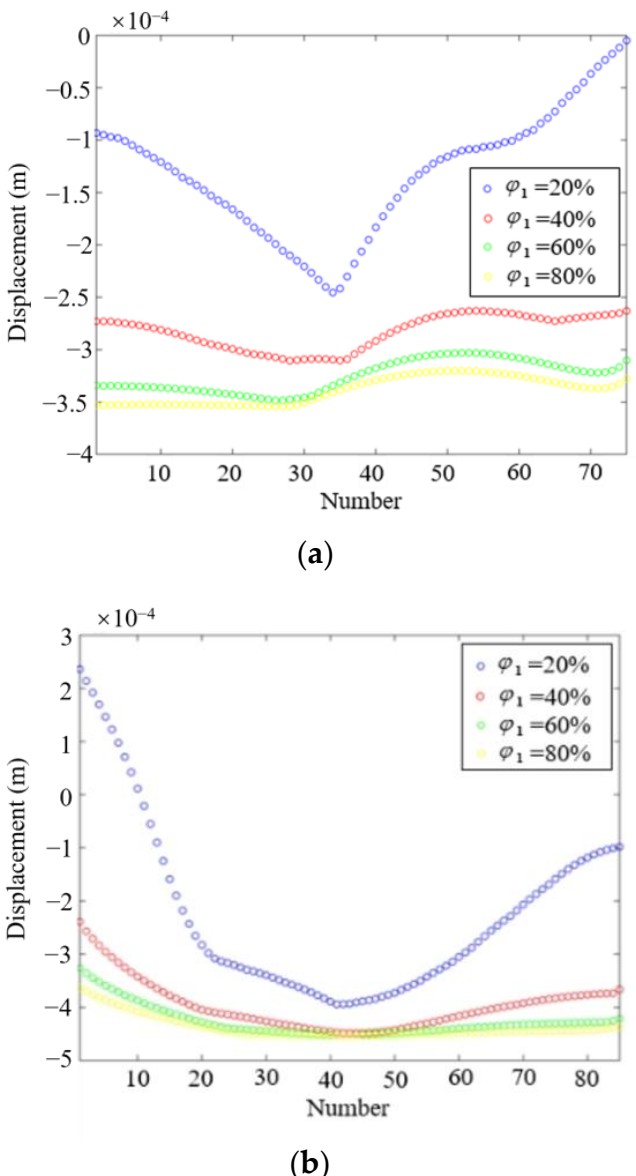

**Figure 8.** Difference in displacement of adjacent disks with numbers of (**a**) LV windings and (**b**) HV windings.

The vertical coordinate represents the difference between the displacements of the two adjacent spacers. More specifically, the longitudinal coordinate indicates the difference in displacement between the lower plane of the upper one and the upper plane of the lower one in the adjacent disks. The ordinate coordinate is denoted by $D$. If $D = 0$, there is no relative displacement between the reference faces representing the adjacent two disks, and the distance between them is the width of the spacer. If $D < 0$, the distance between the reference faces representing adjacent disks is less than the width of the spacer. That is, the spacer is compressed, the disk is in close contact with the spacer, and there is no gap between them. When $D > 0$, the distance between the reference surfaces representing adjacent disks is greater than the width of the spacer, i.e., a gap is created between the disks and the spacer, which has the risk of winding instability.

From Figure 8a, the longitudinal coordinates are all less than zero, which means that the disks and spacers are in close contact, and the stability of the LV windings is satisfactory. From Figure 8b, the gaps appear between the top and bottom disks and spacers of HV windings. The end disks are prone to instability, while the middle disks have good stability. Moreover, the stability of both LV and HV windings changes most when $\varphi_1$ varies from

20% to 40%. In summary, the area proportion of spacers $\varphi_1$ boosts the stability of both LV and HV windings; the vibration stability of LV windings is better than that of HV windings.

In the design of the short-circuit resistance of the winding, an appropriate $\varphi_1$ is essential to avoid the situation of winding instability. Moreover, the fatigue life of the insulation layers of the windings near the ends under continuing transient shock loads should be properly strengthened, so that the breakage due to the frequent collision between the end disks and the spacers can be reduced. Possible measures may be adjustments to the material or thickness.

### 3.3.2. Effect of Axial Preload on Axial Vibration

The axial preload is an essential parameter in the study of winding short-circuit processes. The effect of axial preload on the axial stability of disks is mainly reflected in changing the elasticity modulus of spacers. The simulation results, obtained by changing the axial preload applied to both ends of the winding for the same winding structure, are shown in Figure 9.

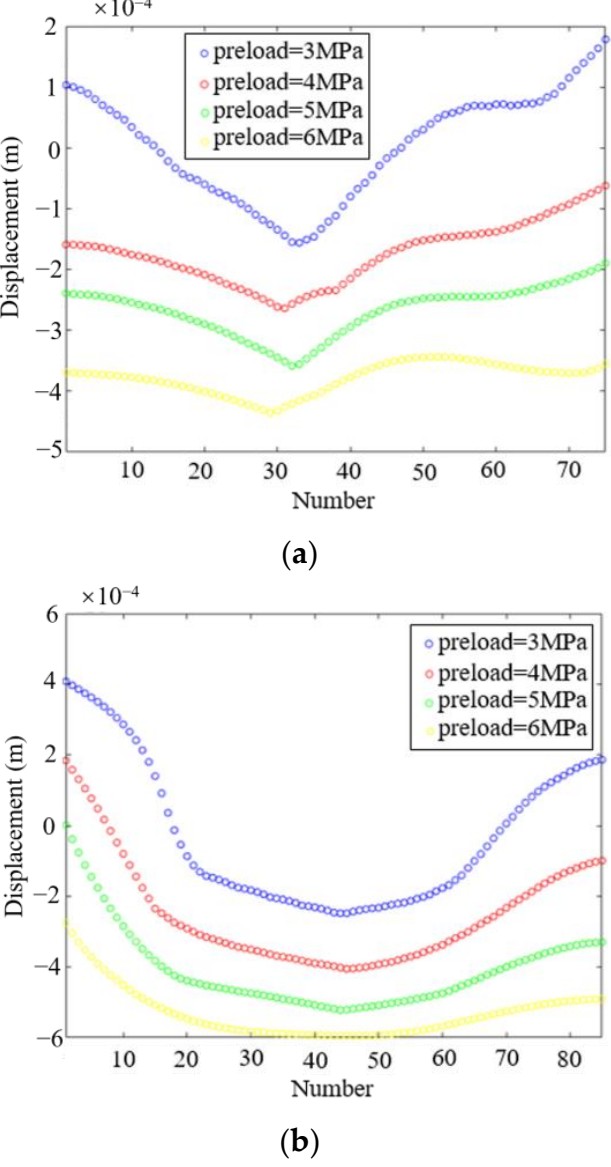

**Figure 9.** Difference in displacement of adjacent disks with numbers of (**a**) LV winding and (**b**) HV winding under 3 MPa, 4 MPa, 5 MPa, and 6 MPa preload.

In Figure 9, the vibration stability of LV and HV windings improved with the growth of the axial preload from 3 MPa to 6 MPa. LV windings are less prone to instability when the axial preload is greater than the value between 3 and 4 MPa, while HV windings undergo stable vibration when the axial preload is greater than 5 MPa. However, high axial preload may cause the windings to tilt or even collapse.

For the transformer model used in this paper, the axial preload should be deliberately selected to ensure the winding axial vibration stability.

### 3.3.3. Vibration and Deformation with Aging Insulated Spacers

The support strength of spacers differs at different aging temperatures. Based on the axisymmetric model of the transformer, the corresponding Young's modulus expressions at different aging temperatures are imported as the material parameters of the model to simulate the axial vibration of the winding at 20 °C, 60 °C, and 130 °C. Comparing the displacement of the winding on each disk at different time instants, the distribution of the maximum value of the axial displacement difference of each disk is obtained, as shown in Figure 10.

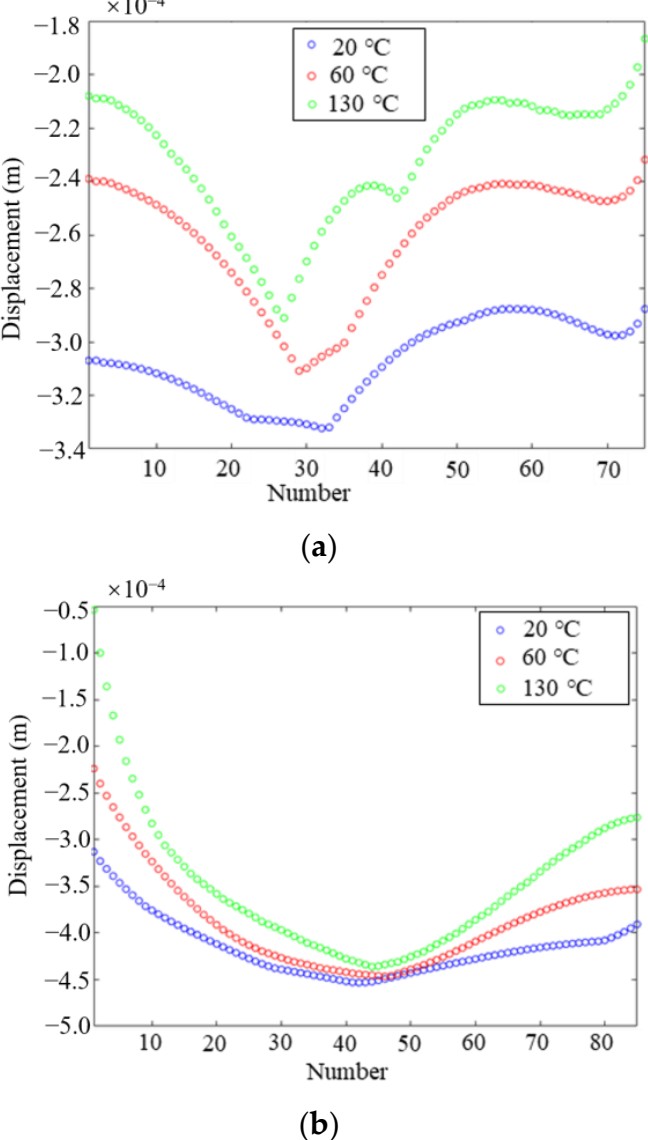

**Figure 10.** Difference in displacement of adjacent disks with numbers of (**a**) LV winding and (**b**) HV winding at 20 °C, 60 °C, and 130 °C.

As seen from Figure 10, the axial displacement differences of the primary and secondary windings of each disk remain negative at all temperatures. It is indicated that the distance between the two disks at each moment in the vibration process is less than the thickness of the insulation spacers. That is, the insulation spacer is always in a compressed state, and the middle spacers of the windings are compressed significantly more than the spacers at both ends of the winding. There is no gap generated between each disk and insulation spacer, and the axial vibration of transformer windings is a stable dynamic process during the short circuit at the above temperature conditions. Overall, the Young's modulus value of the insulation material decreases gradually with the increase in aging temperature, and the degree of the pad is gradually compressed and reduced, which demonstrates that the axial stability of the winding degrades gradually. The simulation results are verified against the experimental conclusions of the existing research.

Using the 3D multilayer model shown in Figure 2b, the winding residual stress distributions under different aging temperatures are obtained when the short-circuit EF amplitude reaches the first wave peak, as shown in Figure 11.

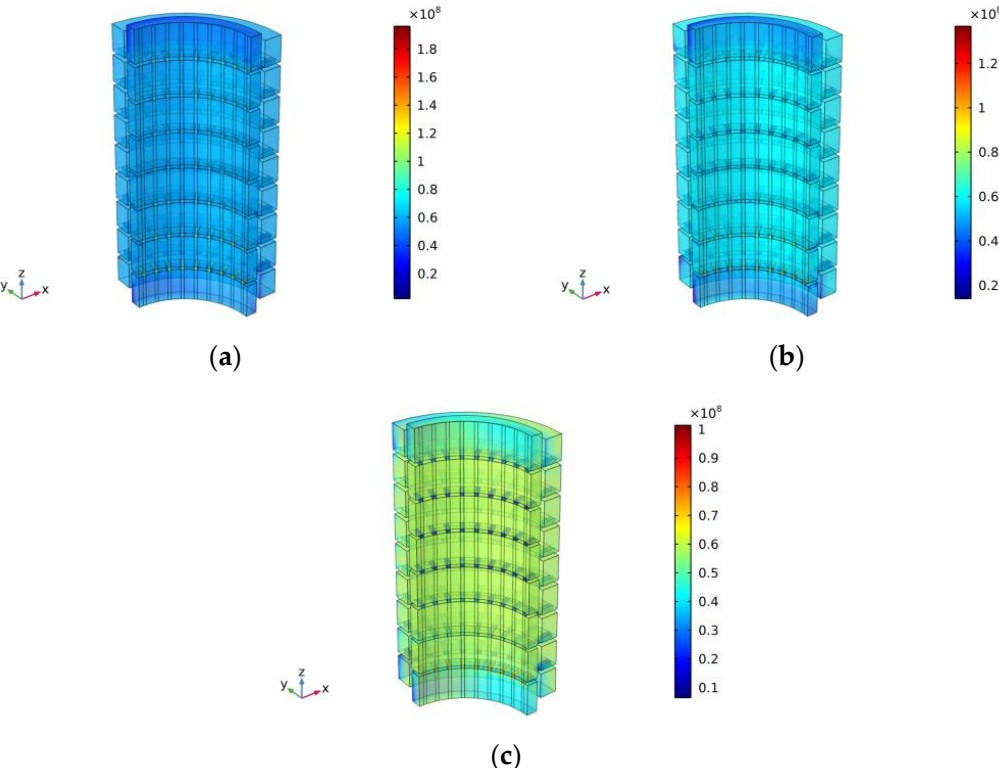

**Figure 11.** Stress distributions of windings under different temperatures: (**a**) 20 °C, (**b**) 60 °C, and (**c**) 130 °C.

In Figure 11, at the maximum amplitude of the short-circuit EF, the stress value of the disks in the middle of the winding is greater than that of the end disks under the radial short-circuit EF. The maximum surface stresses of the primary and secondary windings at 20 °C, 60 °C, and 130 °C are $1.698 \times 10^8$ N/m$^2$, $1.499 \times 10^8$ N/m$^2$, and $1.116 \times 10^8$ N/m$^2$, respectively, and they are all distributed on the primary winding near the support parts. As the temperature increases, the Young's modulus of the insulating paperboard decreases, the elastic support stiffness of the support components to the winding diminishes, and the maximum stress value on the surface of the winding drops gradually. The key to the determination of the spoke destabilization lies in the critical stress value which is equivalent to the same short-circuit EF. The maximum stress value that the winding can withstand decreases gradually as the aging temperature rises, and the transformer is more prone to radial destabilization accidents.

### 3.4. Cumulative Effect of Plastic Deformation

#### 3.4.1. Mechanical Properties of Disks and Spacers

Once the internal mechanical stress value of the windings exceeds the yield strength of the copper conductor, the windings of the transformer produce a slight plastic deformation in the process of the short-circuit fault. In terms of multiple continuous short-circuit currents, eventually, the gradually accumulated plastic deformation leads to winding deformation, dislocation, and other accidents.

The stress–strain curve of the copper conductor in the transformer winding, as a plastic material, is shown in Figure 12. When a short-circuit fault occurs, the copper conductor enters the elastic deformation stage during the destruction of the winding and begins to produce plastic strain after reaching the yield limit, and finally, the plastic deformation accumulates gradually under multiple short-circuit force impacts.

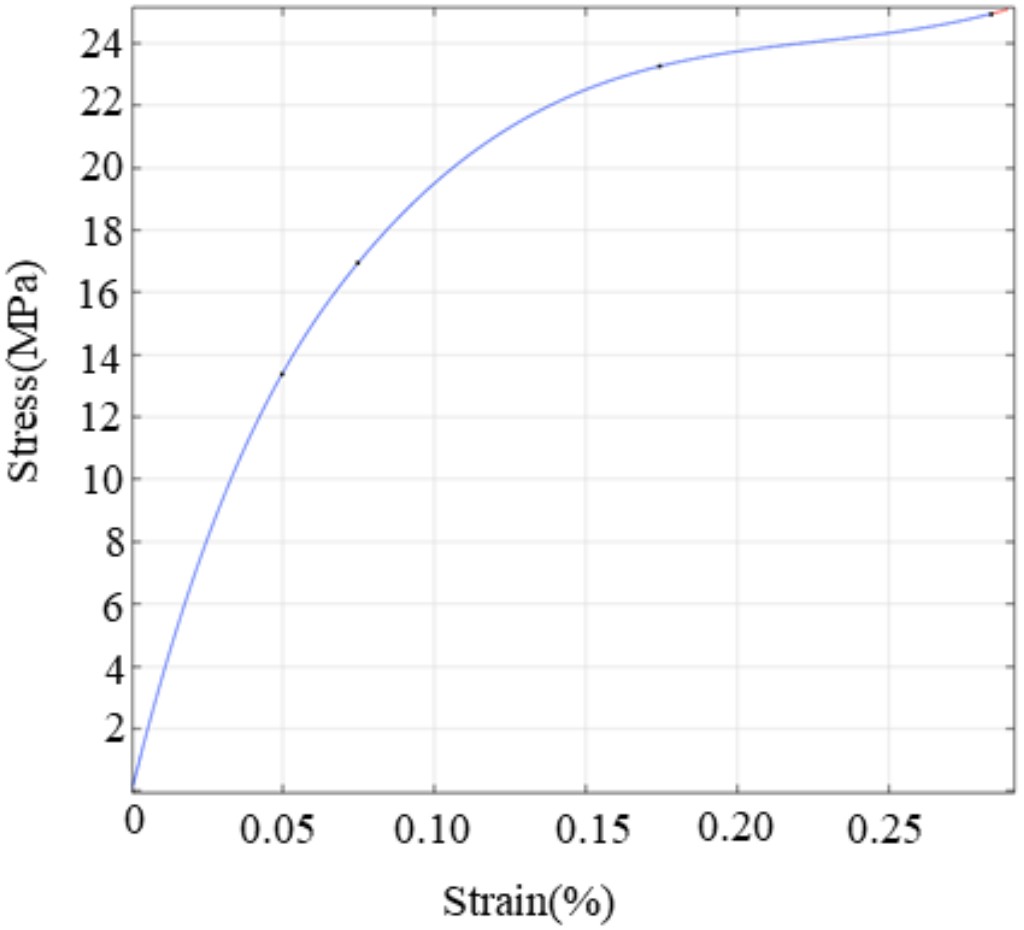

**Figure 12.** Stress–strain curve of the disk.

The spacers are made of nonlinear materials, which means that the dependence of the stress $\sigma$ on the strain $\varepsilon$ exhibits a nonlinear behavior. For the insulating spacer, the $\sigma - \varepsilon$ expression within a stress of 1000 MPa can be expressed by Equation (14), and the variation of the elasticity modulus with strain is shown in Equation (15) [18].

$$\sigma = a\varepsilon + b\varepsilon^3 \tag{14}$$

$$E = \frac{\mathrm{d}\sigma}{\mathrm{d}\varepsilon} = a + 3b\varepsilon^2 \tag{15}$$

where *a* is a constant and *b* is the hardening coefficient. Typically, *a* is 105 MPa and *b* is 1750 MPa. It is clear from Equation (15) that the elasticity modulus *E* of this material depends on strain. In addition, the relationship between the elasticity modulus and strain can be visualized in Figure 13.

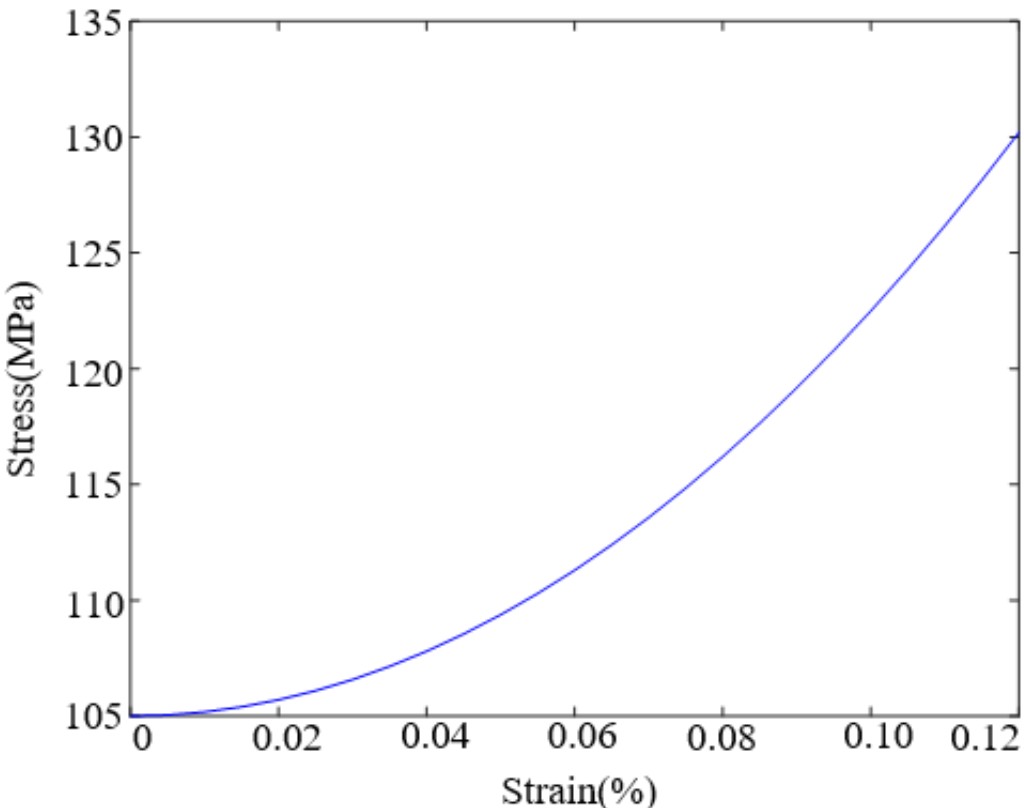

**Figure 13.** Stress–strain curve of the spacer.

Equivalent plastic strain and residual stress are used to represent the strain and stress inside the windings, and a single variable is used to express the degree of plastic deformation. Once the plastic strain is generated, residual stresses are caused by the uneven plastic deformation inside the windings. The residual stress is the internal stress, that remains in the windings after the unloading of the external short-circuit electromagnetism force, and it is in self-balance. When the residual stress and the mechanical stress caused by the external forces superimposed on each other exceed the strength limit of the material, it is likely to cause warping, cracking, or twisting deformations of the structure. If the loading and unloading process of short-circuit EF inside the lined cake generates a different degree of residual stress, it will seriously endanger the stable operation of transformer.

3.4.2. Cumulative Effect of Plastic Strain under Single Short-Circuit Shock

The duration of a short-circuit shock is from *t* = 0 s to *t* = 0.1 s after the short circuit occurs. During a single short-circuit shock, each wave of short-circuit EF may cause plastic deformation of the copper winding. The calculated results of plastic strain and residual stress are shown in Figures 14 and 15 for multiple wave-fronts under a single short-circuit shock, respectively, mainly including the *t* = 0.01 s instant when the short-circuit EF amplitude reaches its maximum and the *t* = 0.1 s instant after the end of the single short-circuit shock.

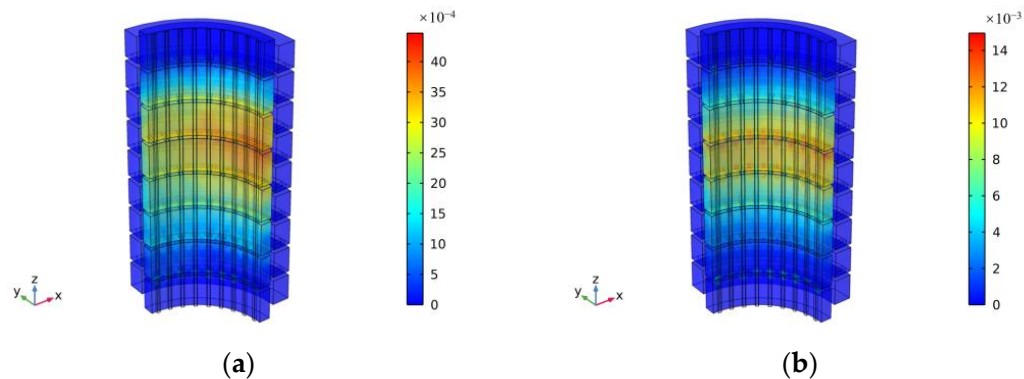

**Figure 14.** Plastic strain distribution under single short-circuit impact at (**a**) *t* = 0.01 s and (**b**) *t* = 0.1 s.

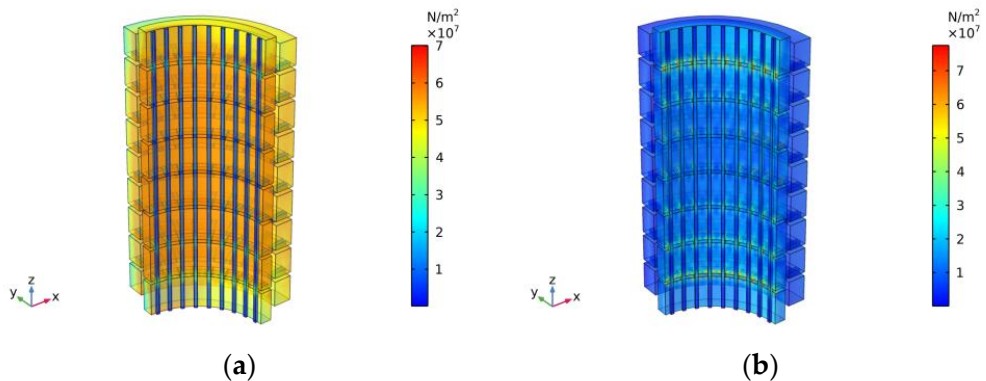

**Figure 15.** Residual stress distribution under single short-circuit shock at (**a**) *t* = 0.01 s and (**b**) *t* = 0.1 s.

As can be seen in Figure 14, the maximum magnitudes of the short-circuit current and short-circuit EF are reached at *t* = 0.01 s, corresponding to a maximum plastic strain of 0.449%. The maximum value of plastic strain increases to 1.571% at *t* = 0.1 s after the end of the single short-circuit shock. The gradual increase in the plastic strain maximum with time shows that multiple wave crests under a single short-circuit shock have a certain cumulative effect. Meanwhile, the results at different time instants indicate that the distribution of plastic strain tends to become more uniform as time advances.

From Figure 15, the maximum value of residual stress increased from $7.446 \times 10^7$ N/m$^2$ to $8.480 \times 10^7$ N/m$^2$ during the duration of a single short-circuit shock, and they all appeared on the disks of the primary winding near the end. The distribution of residual stresses is similar to the distribution of magnetic field leakage under short-circuit conditions as well as the distribution of short-circuit EF.

In the radial direction, the residual stress is concentrated between the two windings near the gap. In the axial direction, it is concentrated in the middle of the windings. The primary winding is subjected to a larger radial short-circuit EF than the secondary winding, so the overall residual stress value is higher than that of the residual stress of the secondary winding. In addition, the minimum value of residual stresses exists and is not zero because of the restraining effect of preload and gravity on the winding in the steady state.

### 3.4.3. Cumulative Effect of Plastic Strain under Multiple Consecutive Short-Circuit Shocks

Due to the multiple reclosures of power systems, recurring short-circuit shocks are a realistic risk to transformers. In this subsection, the stress–strain results obtained from a single short-circuit shock are used as the initial values for another short-circuit shock, and the cumulative effect of the winding under five successive short-circuit shocks is calculated. The distribution of winding plastic strain and residual stress after multiple successive short-circuit shocks is shown in Figure 15, and the results at *t* = 0.01 s after the fifth short-circuit shock are used as an example for comparative analysis.

Comparing the results in Figure 14a with those in Figure 16a, the plastic strain maximum increases by 1.601% after several short-circuit shocks, which is not negligible. Similarly, comparing the results in Figure 15a with those in Figure 16b, the maximum value of residual stress in the winding at $t = 0.01$ s after five short-circuit shocks increases to $1.088 \times 10^8$ N/m². The minimum value also increases, which indicates that the initial residual stress value obtained from the steady-state calculation is related not only to the initial state force, such as the preload force, but also has a certain cumulative effect in the presence of successive short-circuit shocks. The distribution of residual stress increases with the number of short-circuit shocks and is influenced by the distribution of short-circuit EFs, and it is approximately the same as that of the first short-circuit shock.

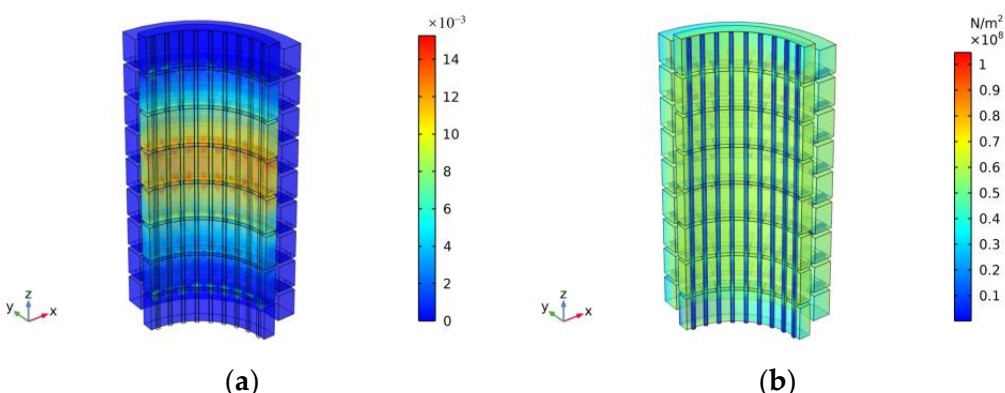

(a)　　　　　　　　　　　　　　　　　(b)

**Figure 16.** The distribution of winding (**a**) plastic strain and (**b**) residual stress after multiple successive short-circuit shocks.

The results of plastic strain and residual stress of winding at different numbers of short-circuit shocks are presented in Figure 17.

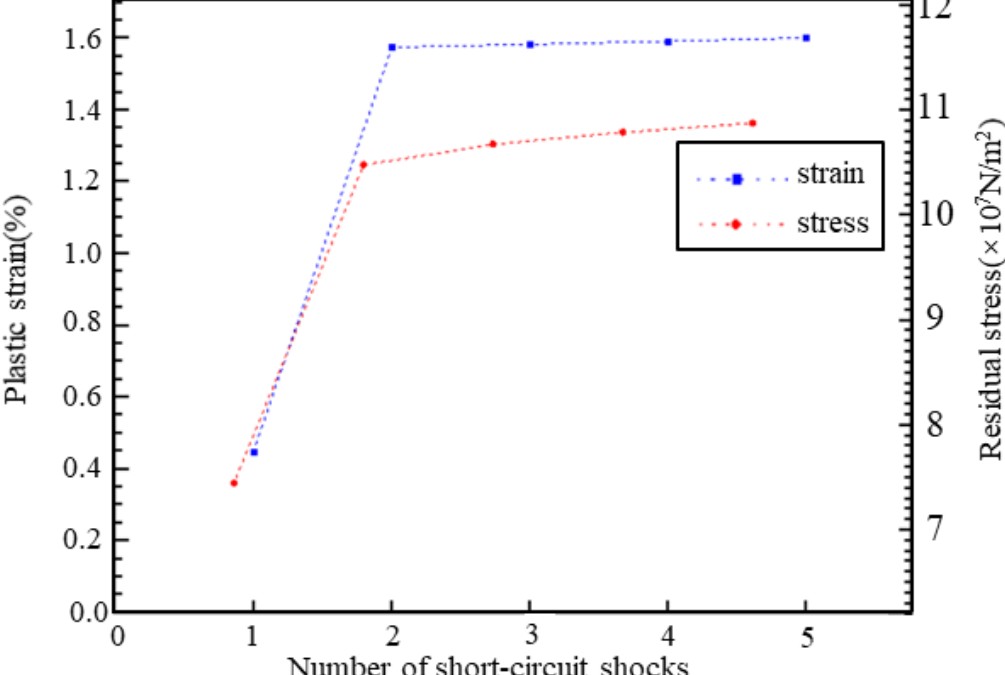

**Figure 17.** The distribution of winding: plastic strain and residual stress for various numbers of short-circuit shocks.

According to the trend of winding plastic strain and residual stress with an increasing number of short-circuit shocks in Figure 17, the residual stress and plastic strain of the winding rises with the increase in the number of short-circuit shocks. The maximum increase in plastic strain and residual stress arises from the second short-circuit shock, and then it accumulates slowly with the increase in the number of shocks.

## 4. Conclusions

The vibration and deformation of transformer windings during short-circuit shocks are generally calibrated statically with the maximum value of the short-circuit current. Compared with most of the existing literature using 2D axisymmetric analysis, in this work, we carry out the dynamic analysis of a 3D model, in which the time evolutions of the short-circuit current and short-circuit electromagnetic force are taken into account. Transient 3D analysis allows flexible geometry and is able to incorporate the nonlinear material properties easily.

Based on our analysis regarding a 110 kV power transformer, it is found that the axial EF tends to compress the winding towards the middle. At the two ends of the windings, the amplitude of axial EF can reach its maximum of $4 \times 10^7$ N, while this value decreases to nearly zero for the middle disks. This is understandable because the radial leakage exhibits a similar distribution, namely, the peak amplitude appears at the two ends.

The axial vibration displacements of the winding under short-circuit conditions with varying pad area ratios and preload forces are investigated. It is concluded that increasing the area proportion of spacers in each layer, from 20% to 40%, can considerably improve the axial vibration stability, and a larger proportion is beneficial to improving the axial vibration stability. The vibration stability of LV and HV windings are also improved with the growth of the axial preload from 3 MPa to 6 MPa. In our example, a preload force of no less than 5 MPa can completely avoid instability.

The influence of aging temperature on the radial deformation of the windings is also analyzed. It is found that the maximum surface stress of the windings drops by 34% when the aging temperature varies from 20 °C to 130 °C, which indicates that excessively high aging temperature leads to potential axial instability accidents.

The nonlinear material properties of the insulating components and disks are taken into account, and the cumulative effects of multiple short-circuit current peaks under a single short-circuit shock as well as multiple consecutive short-circuit shocks are illustrated. It is shown that the plastic strain and residual stress dramatically increase by roughly 300% and 40.7% when the second short-circuit shock occurs, respectively, then they grow very slowly under the following shocks.

**Author Contributions:** Conceptualization, J.W., Y.X., X.M. and L.Y.; methodology, J.W., Y.X., X.M. and L.Y.; software, Y.X. and Z.Z.; validation, J.W., Y.X., X.M. and Z.Z.; data curation, Y.X. and Z.Z.; writing—original draft preparation, J.W., Y.X. and X.M.; supervision, X.M.; project administration, X.M. All authors have read and agreed to the published version of the manuscript.

**Funding:** This research was supported by State Grid Anhui Electric Power Co., Ltd. Electric Power Science Research Institute, grant number SGAHDK00SPJS2000312, and in part by the Natural Science Basic Research Program of Shaanxi, program number 2022JQ-484.

**Data Availability Statement:** Data are available upon reasonable request to the corresponding author.

**Conflicts of Interest:** The authors declare no conflict of interest.

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
