# Peer review of "Numerical Investigations for Vibration and Deformation of Power Transformer Windings under Short-Circuit Condition"

_energies, doi:10.3390/en16145318_

Round 1

Reviewer 1 Report

Dear authors, 

The paper is about finite element analysis of a 63 MVA transformer in a short circuit situation, aiming to determine points of vibration and deformation. Although one can see an extensive work, the paper does not have a clear contribution nor an important contextualization, bibliographical review. The paper in its actual format seems a technical report. So, an extensive review must be done to be reconsidered for publication. 

1) Provide some background formulation for the vibration and deformation that occurs due to short circuits. By doing so, the methodology could be applied to any transformer, and not only to the presented case study.   

2) I would suggest to include a section about the finite element analysis principles to help the reader understand the numerical technique applied to the case study.   

3) Another section that should be included would be about the case study of the 63 MVA transformer. Move table 1 to this section. When it is included in the introduction section, the paper sounds more like a technical report, as stated in the beginning.   

4) Figure 8, 11-13 - what are the units of y axis? Make it clear in the figures.   

5) Rewrite the paper showing, at first, the theoretical background. Then, present the case study with all results, making it an example of the usage of the theory presented before.

Author Response

Dear Editor and Reviewers,

Thank you very much for your letter and comments on our manuscript entitled “Numerical investigations for Vibration and Deformation of Power Transformer Windings under Short-circuit Condition” (ID Energies-2408672). These comments are very valuable and helpful for improving our paper. We have studied all of these comments very carefully and made the corresponding corrections which we hope meet with your approval. The main corrections in the revised manuscript and the detailed responses to the reviewers’ comments are listed as follows:

Comments to the Author

The paper is about finite element analysis of a 63 MVA transformer in a short circuit situation, aiming to determine points of vibration and deformation. Although one can see an extensive work, the paper does not have a clear contribution nor an important contextualization, bibliographical review. The paper in its actual format seems a technical report. So, an extensive review must be done to be reconsidered for publication. 

1) Provide some background formulation for the vibration and deformation that occurs due to short circuits. By doing so, the methodology could be applied to any transformer, and not only to the presented case study.   

Response to this comment:

Thank you very much for your comments. We agree that the original manuscript is not structured properly and essential formulations are missed. The newly added section 2 entitled “Theory and Formulations” contain background formulations of our research.

2) I would suggest to include a section about the finite element analysis principles to help the reader understand the numerical technique applied to the case study.   

Thank you very much for your advice. We have added section 2.3 to clarify the adopted FEM software, time-stepping scheme, and discretized FEM formulation in a compact matrix form. Further technical details, e.g., the shape functions, evaluation of the mass- and stiffness- matrices, are omitted because it’s not our focus. We have added a reference for further details, which is the classic text book by O. C. Zienkiewicz.

3) Another section that should be included would be about the case study of the 63 MVA transformer. Move table 1 to this section. When it is included in the introduction section, the paper sounds more like a technical report, as stated in the beginning.

Thank you very much for your advice. The revised manuscript has been restructured.

4) Figure 8, 11-13 - what are the units of y axis? Make it clear in the figures.

We apologize for our carelessness. These figures have been updated. It’s worth noting that for some figures, y axis corresponding to strain is unitless.

5) Rewrite the paper showing, at first, the theoretical background. Then, present the case study with all results, making it an example of the usage of the theory presented before.

Thank you for this comment. The revised manuscript has been restructured.

In a few words, we have tried our best to improve the manuscript and make changes in the revised version of the paper according to all the reviewers’ good comments. We greatly appreciate the Editors/Reviewers’ warm work and hope that the corrections will meet with approval. We look forward to hearing from you about our revised paper.

With best regards,

Jiawei Wang

School of Electrical Engineering, Xi’an Jiaotong University

Xi’an 710049, P. R. China

Reviewer 2 Report

A very well-structured manuscript focused on the vibration and deformation of power transformer windings. Congrats to the authors.

State of the art is relating the nowadays solution and the research interest area, also the references are the majority of the past 10 years which reveals the up-to-date research.

Good simulation support for the transformer model proposed.

The work is based on a very well-done expert interpretation of the resulting data.

The model proposed can be useful for future researchers in order to anticipate the deformation of electrical machines due to short circuit currents.

Good mathematical support for the simulation proposed. Anyway for the deformation analysis is not that clear what are the short circuit current parameters (maximum value reached, or distance of occurrence…) In the presented model proposed a step-up transformer, which usually is three phase construction, do the other phases influence in any manner the analyzed phase windings? Of course from the deformation point of view… .

The paper is interesting and can be published in the journal with minor changes, regarding article structure form according to the journal template.

Regards to the authors,

Reviewer

Author Response

Dear Editor and Reviewers,

Thank you very much for your letter and comments on our manuscript entitled “Numerical investigations for Vibration and Deformation of Power Transformer Windings under Short-circuit Condition” (ID Energies-2408672). These comments are very valuable and helpful for improving our paper. We have studied all of these comments very carefully and made the corresponding corrections which we hope meet with your approval. The revised manuscript is the file named “Manuscript (revision)”. The main corrections in the revised manuscript and the detailed responses to the reviewers’ comments are listed as follows:

Reviewer: 2

Comments to the Author

A very well-structured manuscript focused on the vibration and deformation of power transformer windings. Congrats to the authors.

State of the art is relating the nowadays solution and the research interest area, also the references are the majority of the past 10 years which reveals the up-to-date research.

Good simulation support for the transformer model proposed.

The work is based on a very well-done expert interpretation of the resulting data.

The model proposed can be useful for future researchers in order to anticipate the deformation of electrical machines due to short circuit currents.

Good mathematical support for the simulation proposed. Anyway for the deformation analysis is not that clear what are the short circuit current parameters (maximum value reached, or distance of occurrence…) In the presented model proposed a step-up transformer, which usually is three phase construction, do the other phases influence in any manner the analyzed phase windings? Of course from the deformation point of view… .

The paper is interesting and can be published in the journal with minor changes, regarding article structure form according to the journal template.

Regards to the authors,

Response to this comment:

Thank you very much for your favorable comments.

Regarding the missing short circuit current parameters, the peak amplitude is A appearing at the time instant t=0.01 s. Detailed transient waveforms can be found in Figure 4 in the revised manuscript.

As for the distance of occurrence, in our modelling the short circuit current flows through the whole windings regardless of the position of fault. The position of the fault mainly affects the effective impedance of the short-circuit model.

Finally, we justify our simplification of the common three-phase transformers as a single-phase model. The force between the adjacent windings of the three-phase transformer is neglected under a sufficient axial preload, thus we only calculate one phase winding. This conclusion is from the newly added reference [17] in the revised manuscript, which we also list here for your convenience.

Reference

  1. Xie Y. Power Transformer Handbook: Second Edition. Machinery Industry Press, Beijing, 2014. (In Chinese)

In a few words, we have tried our best to improve the manuscript and make changes in the revised version of the paper according to all the reviewers’ good comments. We greatly appreciate the Editors/Reviewers’ warm work and hope that the corrections will meet with approval. We look forward to hearing from you about our revised paper.

With best regards,

Jiawei Wang

School of Electrical Engineering, Xi’an Jiaotong University

Xi’an 710049, P. R. China

Reviewer 3 Report

COMMENTS:

1. Present in figure 2 an extra figure to observe a longer period of time, for example, 0.3 seconds.

2. Figure 3 does not present units, the color bar.

3. The coordinates are not appreciated correctly, change the coordinate plane to its proper shape.

4. Image 3 should be improved, it is not very clear, I suggest making it bigger and improving its quality.

5. Improve the wording from line 68 to line 69. It is understandable for a specific audience but it should be placed in a more digestible way for a broader audience.

6. Add an image where the variables of equation 2 can be seen.

7. Line 144, mentions that it should be reinforced, but how much should be reinforced…double…triple?

8. Improve the quality and distribution of figure 4.

9. Reference of equation 3 and equation 4.

10. Add a table that concretely explains the most significant results.

11. Add a discussion of the results of this work and other previous works presented in the literature.

12. In the conclusions, results are given as smaller, smaller, larger. Changing these results for measurable results, for example in percentages, could be an option.

13. Place the units to figure 11, 12.

14. Equation 5 presents a key, which indicates this parenthesis, apparently something was forgotten to be placed or the parenthesis is just left over.

15. Place the units in figure 13.

16. The conclusions must be improved.

Author Response

Dear Editor and Reviewers,

Thank you very much for your letter and comments on our manuscript entitled “Numerical investigations for Vibration and Deformation of Power Transformer Windings under Short-circuit Condition” (ID Energies-2408672). These comments are very valuable and helpful for improving our paper. We have studied all of these comments very carefully and made the corresponding corrections which we hope meet with your approval. The revised manuscript is the file named “Manuscript (revision)”. The main corrections in the revised manuscript and the detailed responses to the reviewers’ comments are listed as follows:

Comments to the Author

  1. Present in figure 2 an extra figure to observe a longer period of time, for example, 0.3 seconds.

Response to this comment:

Thank you very much for your advice. We have extended the total time of figure 2, see figure 4 in the revised manuscript.

  1. Figure 3 does not present units, the color bar.
  2. The coordinates are not appreciated correctly, change the coordinate plane to its proper shape.
  1. Image 3 should be improved, it is not very clear, I suggest making it bigger and improving its quality.

Response to these comments:

We apologize for our carelessness. All these issues have been modified.

  1. Improve the wording from line 68 to line 69. It is understandable for a specific audience but it should be placed in a more digestible way for a broader audience.

Response to this comment:

Thanks for pointing this out. The wording is modified as “In Figure 1 (a), a simplified axisymmetric model of the transformer is presented, which is obtained by neglecting some unsymmetric geometry details.”

  1. Add an image where the variables of equation 2 can be seen.

Response to this comment:

Thanks for your advice. Please refer to Figure 7 in the revised manuscript for corresponding update.

  1. Line 144, mentions that it should be reinforced, but how much should be reinforced…double…triple?

Response to this comment:

Thanks for this comment. We agree that quantitative expressions would be more favorable. However, it requires expertise in material science to make such an accurate conclusion, which is beyond our knowledge. We have modified wording here to clarify that the ultimate goal is to improve the fatigue life of the insulation layers under continuing transient shock loads. To achieve such goal, you can either increase the thickness or alter the material of certain layers.

  1. Improve the quality and distribution of figure 4.

Response to this comment:

Thanks for your advice. Figure 4 has been updated, see figure 6 in the revised manuscript.

  1. Reference of equation 3 and equation 4.

Response to this comment:

Thanks for your advice. Please refer to reference [18] in the revised manuscript.

  1. Add a table that concretely explains the most significant results.

Response to this comment:

Thanks for your advice. Please refer to Table 2 in the conclusion section of the revised manuscript.

  1. Add a discussion of the results of this work and other previous works presented in the literature..

Response to this comment:

Thanks for your advice. Please refer to the first paragraph in the conclusion section of the revised manuscript.

  1. In the conclusions, results are given as smaller, smaller, larger. Changing these results for measurable results, for example in percentages, could be an option.

Response to this comment:

Thanks for your advice. Please check the revised conclusion section, where we have incorporated more quantitative conclusions.

  1. Place the units to figure 11, 12.

Response to this comment:

Thanks for reminding. All these issues have been modified.

  1. Equation 5 presents a key, which indicates this parenthesis, apparently something was forgotten to be placed or the parenthesis is just left over.

Response to this comment:

Thanks for your comment. We have verified that the parenthesis is indeed just left over, indicating that the three equations together describe the dependence of stress on the applied force load.

  1. Place the units in figure 13.

Response to this comment:

Thanks for reminding. Figure 13 has been updated, see figure 15 in the revised manuscript.

  1. The conclusions must be improved.

Response to this comment:

Thanks for your advice. We have followed your suggestions and modified the conclusion section accordingly.

In a few words, we have tried our best to improve the manuscript and make changes in the revised version of the paper according to all the reviewers’ good comments. We greatly appreciate the Editors/Reviewers’ warm work and hope that the corrections will meet with approval. We look forward to hearing from you about our revised paper.

With best regards,

Jiawei Wang

School of Electrical Engineering, Xi’an Jiaotong University

Xi’an 710049, P. R. China

Round 2

Reviewer 1 Report

Dear authors, 

a clear improvement has been made from the first version of the paper. 

Author Response

Thank your very much for your detailed comments for improving our work, and the approval of our current manuscript.